# Effect of METTL3 Gene on Lipopolysaccharide Induced Damage to Primary Small Intestinal Epithelial Cells in Sheep

**DOI:** 10.3390/ijms25179316

**Published:** 2024-08-28

**Authors:** Yanjun Duan, Xiaoyang Lv, Xiukai Cao, Wei Sun

**Affiliations:** 1College of Veterinary Medicine, Yangzhou University, Yangzhou 225009, China; 2Joint International Research Laboratory of Agriculture and Agri-Product Safety of Ministry of Education of China, Yangzhou University, Yangzhou 225009, China; 3International Joint Research Laboratory in Universities of Jiangsu Province of China for Domestic Animal Germplasm Resources and Genetic Improvement, Yangzhou University, Yangzhou 225009, China; 4College of Animal Science and Technology, Yangzhou University, Yangzhou 225009, China

**Keywords:** LPS, IECs, METTL3, inflammatory response, apoptosis, oxidative stress

## Abstract

Newborn lambs are susceptible to pathogenic bacterial infections leading to enteritis, which affects their growth and development and causes losses in sheep production. It has been reported that N6-methyladenosine (m6A) is closely related to innate immunity, but the effect of m6A on sheep small intestinal epithelial cells (IECs) and the mechanism involved have not been elucidated. Here, we investigated the effects of m6A on lipopolysaccharide (LPS)-induced inflammatory responses, apoptosis and oxidative stress in primary sheep IECs. First, the extracted IECs were identified by immunofluorescence using the epithelial cell signature protein cytokeratin 18 (*CK18*), and the cellular activity of IECs induced by different concentrations of LPS was determined by the CCK8 assay. Meanwhile, LPS could induce the upregulation of mRNA and protein levels of IECs cytokines *IL1β*, *IL6* and *TNFα* and the apoptosis marker genes *caspase-3*, *caspase-9*, *Bax*, and apoptosis rate, reactive oxygen species (ROS) levels and mRNA levels of *CAT*, *Mn-SOD* and *CuZn-SOD*, and METTL3 were found to be upregulated during induction. It was hypothesized that *METTL3* may have a potential effect on the induction of IECs by LPS. Overexpression and knockdown of *METTL3* in IECs revealed that a low-level expression of METTL3 could reduce the inflammatory response, apoptosis and ROS levels in LPS-induced IECs to some extent. The results suggest that *METTL3* may be a genetic marker for potential resistance to cellular damage.

## 1. Introduction

Harmful bacteria and microorganisms, such as *F17 Escherichia coli* [1], and *Clostridium perfringens* [2], cause enteric diseases in sheep, including diarrhea, enterotoxemia, and other symptoms that severely impact the sheep industry. LPS, a major component of the cell wall of Gram-negative bacteria, possesses high immunogenicity, cytotoxicity, and immune cell activation properties [3,4]. IECs are crucial for maintaining the intestinal barrier function and play a pivotal role in resisting pathogenic bacterial infections by blocking their invasion while also receiving signals from the gut environment [5].

It is widely acknowledged that LPS acts as an activator of Toll-like receptor 4 (TLR4). As a pattern recognition receptor, TLR4 can identify pathogen-associated molecular patterns, such as bacterial LPS and viral structural proteins, to activate innate immunity [6]. Upon TLR4 activation, host responses are modulated through MYD88-dependent and MYD88-independent pathways. In the MYD88-dependent pathway, a complex is formed between MYD88 and members of the IRAK kinase family, leading to the subsequent activation of NF-κB and MAPK signaling pathways via TRAF6 and TAK1 signal transduction processes. This triggers inflammatory responses, oxidative stress, and cellular apoptosis [7,8,9,10]. In contrast, in the MYD88-independent pathway, activation of TRIF and TRAF3 recruits IKKε/TBK1, leading to IRF3 phosphorylation and resulting in beta interferon expression, which affects host antibacterial immunity [11]. Numerous studies have reported that LPS influences cellular inflammatory response mechanisms along with apoptosis and oxidative stress through TLR4. For instance, LPS stimulates inflammation in bovine endometrial epithelial cells (bEECs) via the TLR4/NF-κB signaling pathway in vitro and similarly induces inflammation during mouse endometritis in vivo [12]. Interfering with TLR4 has been shown to reduce the inflammatory response induced by LPS through the NF-κB pathway modulation, while also decreasing apoptosis triggered by the P53 pathway induction in IPEC-J2 cells [13]. Additionally, in bovine rumen epithelial cells, LPS induces a cellular inflammatory response alongside oxidative stress through TLR4-NF-κB signaling pathway activation [14]. Epithelial cells represent primary infection sites for harmful intestinal bacteria within the intestine. However, there are limited reports on bacterial infection of intestinal epithelial cells in sheep IECs. Hence, it is imperative to understand the inductive effect of LPS on IECs.

N6-methyladenosine (m6A) is the most prevalent chemical modification of the eukaryotic messenger RNA (mRNA) and plays a pivotal role in diverse biological processes. This methylation modification of m6A has been demonstrated to be reversible through the involvement of methyltransferases (Writers), demethylase enzymes (Erasers), and m6A-binding proteins (Readers). Notably, methyltransferases such as methyltransferase 3 (*METTL3*), methyltransferase 14 *(METTL14*), and the WT1-associated protein (*WTAP*) are capable of catalyzing the m6A modification on mRNA adenylate. Demethylase enzymes encompass alpha-ketoglutarate dependent dioxygenase (*FTO*) and alkB homolog 5 (*ALKBH5*), which facilitate the removal of m6A modifications from bases. The reader proteins include YTHDF family members (YTHDFs) and IGF2BP family members (IGF2BPs), whose function involves the recognition of m6A-modified bases, thereby influencing gene transcription, degradation, and translation [15,16,17]. It has been well established that m6A is intricately involved in innate immunity, given its capacity to influence cytokine alterations and activators affecting dendritic cells [18]. However, there is currently no report on the involvement of m6A in immune responses within sheep IECs.

This study found that a certain concentration of LPS induced IECs, leading to an increase in METTL3 protein expression. It is speculated that METTL3 is related to LPS induction. The function of METTL3 in LPS-induced damage to sheep IECs was further explored, providing a new perspective for breeding sheep with disease resistance.

## 2. Results

### 2.1. Effects of Different Concentrations of LPS on IECs Activity and METTL3 Protein Expression Levels

Immunofluorescence analysis (Figure 1A) was performed after isolating the extracted IECs. The expression of the epithelial cell marker CK18 confirmed that the IECs achieved a high level of purity, making them suitable for subsequent experiments. IECs were treated with LPS at concentrations of 0 μg/mL, 0.5 μg/mL, 1 μg/mL, 2 μg/mL, 5 μg/mL, and 10 μg/mL for 4 h. Compared to the control group, 0.5 μg/mL, 1 μg/mL, and 2 μg/mL LPS significantly enhanced IEC viability, while 5 μg/mL and 10 μg/mL LPS significantly reduced cell viability (Figure 1B). Additionally, LPS stimulation at 2 μg/mL led to an increase in METTL3 protein expression in IECs (Figure 1C).

### 2.2. LPS Induces Inflammatory Response, Apoptosis, and Oxidative Stress in IECs

According to the results of the CCK8 experiment (Figure 1B), the cell survival rate began to decline at an LPS concentration of 5 μg/mL, leading to the selection of 5 μg/mL as the subsequent LPS concentration. In the assessment of the inflammatory response, qRT-PCR analysis revealed that the mRNA levels of the cytokines *IL1β*, *IL6*, and *TNFα* were significantly elevated following LPS induction (Figure 2A–C). Additionally, ELISA analysis demonstrated that under various concentrations of LPS stimulation, the protein levels of IL1β, IL6, and TNFα were also significantly upregulated in cell cultures (Figure 2D–F). Analysis of cell apoptosis indicated that qRT-PCR results showed a significant increase in the mRNA levels of apoptosis markers *caspase-3*, *caspase-9*, and *Bax* following LPS induction (Figure 2G–I). Furthermore, flow cytometry results indicated that the apoptosis rates were significantly higher in the LPS group compared to the control group (Figure 2J,K). In the oxidative stress assessment, ROS levels were significantly elevated in the LPS group (Figure 2L), and qRT-PCR results demonstrated that the mRNA levels of antioxidant marker genes *CAT*, *Mn-SOD*, and *CuZn-SOD* were significantly reduced in the LPS group (Figure 2M–O). In summary, LPS induces inflammatory responses, apoptosis, and oxidative stress in IECs.

### 2.3. Efficiency Detection of Overexpression and Interference of METTL3

The *METTL3* overexpression vector was constructed using the pcDNA3.1 empty vector (Figure 3A), followed by double enzyme digestion and ligation. The successful construction of the METTL3 overexpression vector was confirmed by PCR (Figure 3B) and Sanger sequencing (Figure 3C), which verified the correct sequence of the recombinant *METTL3* overexpression vector.

To evaluate the transfection efficiency of the *METTL3* overexpression vector and interference sequence, the pcDNA3.1-EGFP fluorescent vector (Figure 4A) and FAM-labeled negative control (Figure 4B) were transfected into IECs. After 36 h, the expression of the fluorescent protein was observed under a fluorescence microscope. The results indicated that both the pcDNA3.1 vector and the interference sequence successfully transfected IECs. qRT-PCR analysis showed that the mRNA levels in the *METTL3* overexpression group (OE-METTL3) were significantly higher than those in the pcDNA3.1 empty vector group (OE-NC) (Figure 4C). The three interference sequences targeting *METTL3* were siR-METTL3-1616, siR-METTL3-199, and siR-METTL3-1176. Among these, the mRNA levels of siR-METTL3-1616 and siR-METTL3-199 were significantly lower than that of siR-NC (Figure 4D), with siR-METTL3-1616 demonstrating the most pronounced effect. Therefore, siR-METTL3-1616 was selected for subsequent experiments. Western blot results indicated that METTL3 protein levels were significantly upregulated following *METTL3* overexpression (Figure 4E) and significantly downregulated after interference (Figure 4F). These findings demonstrate that the efficiencies of *METTL3* overexpression and interference are satisfactory, thereby fulfilling the requirements for subsequent experiments.

### 2.4. METTL3 Promotes LPS-Induced Inflammatory Response in IECs

To evaluate the impact of METTL3 on the inflammatory response in LPS-induced IECs, qRT-PCR was conducted to analyze the mRNA levels of cytokines *IL1β*, *IL6*, and *TNFα* after LPS-induced *METTL3* overexpression and interference. The results showed that the mRNA levels of IL1β, IL6, and TNFα were significantly upregulated following *METTL3* overexpression (Figure 5A–C). Conversely, the mRNA levels of *IL1β* and *IL6* were significantly downregulated after *METTL3* interference, while the mRNA levels of *TNFα* showed no significant difference with the treatments (Figure 5D–F). Additionally, ELISA was used to measure the protein levels of IL1β, IL6, and TNFα. After *METTL3* overexpression, the protein levels of these cytokines were significantly increased in the cell culture (Figure 5G–I). In contrast, after *METTL3* interference, the protein levels of IL1β and IL6 were significantly downregulated, with no significant difference observed in the protein level of TNFα (Figure 5J–L). These results suggest that METTL3 promotes the LPS-induced inflammatory response in IECs.

### 2.5. METTL3 Promotes LPS-Induced Apoptosis in IECs

The qRT-PCR results showed that overexpressing *METTL3* in LPS-induced IECs significantly upregulated the apoptosis marker genes *Caspase-3*, *Caspase-9*, and *Bax* (Figure 6A–C). Conversely, after LPS induction and interference with *METTL3* expression in IECs, the apoptosis marker genes *Caspase-3*, *Caspase-9*, and *Bax* were significantly downregulated (Figure 6D–F). Additionally, flow cytometry results demonstrated that in LPS-induced IECs overexpressing *METTL3*, the cell apoptosis rate was significantly increased (Figure 6G,H). In contrast, after LPS induction in IECs with *METTL3* interference, the cell apoptosis rate was significantly decreased (Figure 6I,J). These results suggest that METTL3 promotes apoptosis in IECs.

### 2.6. METTL3 Promotes LPS-Induced Oxidative Stress and Inhibits Antioxidant Activity in IECs

Analysis of flow cytometry results showed that ROS levels in IECs were significantly increased after LPS-induced overexpression of *METTL3* (Figure 7A), while the mRNA levels of antioxidant marker genes *CAT*, *Mn-SOD*, and *CuZn-SOD* were significantly downregulated (Figure 7B–D). Conversely, there was no significant difference in ROS levels in IECs after LPS-induced interference with *METTL3* (Figure 7E). However, the mRNA levels of antioxidant marker genes *CAT* and *Mn-SOD* were significantly increased, with no significant difference observed in *CuZn-SOD* mRNA levels (Figure 7F–H). These results suggest that METTL3 promotes oxidative stress and inhibits the antioxidant response in IECs.

## 3. Discussion

There is growing evidence that inflammatory responses, oxidative stress, and apoptosis are three key mechanisms underlying the onset and progression of many diseases [19,20,21]. m6A modification plays a role in various biological processes, and this process is reversible, maintaining a dynamic balance. METTL3, as a writer, can alter the level of m6A methylation through overexpression or knockdown, thereby affecting mRNA stability, translation, and other functions [22]. In this study, we investigated the effects of METTL3 on LPS-induced inflammatory responses, oxidative stress, and apoptosis in sheep IECs.

In the study of LPS-induced inflammatory responses, METTL3 protein levels were found to be upregulated in sheep IECs, which in turn triggered inflammatory responses. Overexpression of *METTL3* promoted the expression levels of cytokines IL1β, IL6, and TNFα, whereas the silencing of METTL3 reversed these effects. These findings suggest that METTL3 plays a crucial role in the LPS-induced inflammatory response in sheep IECs. In a study examining the role of METTL3 in LPS-induced inflammatory responses in mouse epithelial cells, METTL3 knockdown was found to enhance cell viability, inhibit apoptosis, and reduce levels of pro-inflammatory cytokines. The study posited that the transcription factor NF-κB played a critical role, although it did not thoroughly explore the underlying regulatory mechanisms [23]. Another study on the impact of METTL3 on the susceptibility of IPEC-J2 cells to *E. coli F18* revealed that METTL3 activates the NF-κB signaling pathway, influencing immune regulation by enhancing the transcription and translation of *IKBKG* in an m6A-YTHDF1-dependent manner [24]. Furthermore, METTL3 silencing was shown to alleviate renal inflammation and programmed cell death induced by LPS stimulation. The mechanism involves TAB3 as the target of METTL3 action, with IGF2BP2 binding to the terminator codon region with m6A modification, enhancing the stability of TAB3, which is an upstream activator of NF-κB and MAPK. Ultimately, this signal transduction leads to renal inflammation and programmed cell death [25]. In LPS-mediated microglial inflammation, METTL3 binds to TRAF6 in an m6A-dependent manner, promoting the activation of the NF-κB pathway and influencing the microglial inflammatory response [26]. These findings indicate that METTL3 can act on multiple targets to affect the inflammatory response, and that low expression of METTL3 can reduce inflammation. However, there are also contrasting findings; for example, overexpression of METTL3 has been shown to attenuate the LPS-induced inflammatory response in macrophages through the NF-κB pathway [27].

Apoptosis is an essential process in various biological phenomena, such as normal cell turnover, immune system regulation, embryonic development, and chemically induced cell death [28]. This study investigates the effect of METTL3 on LPS-induced apoptosis in sheep IECs. The results show that overexpression of METTL3 promotes apoptosis in IECs, whereas interference with METTL3 inhibits apoptosis in IECs, indicating that METTL3 is a regulatory factor affecting apoptosis in IECs. METTL3 was expressed at high levels in both pediatric pneumonia patients and cellular models, where downregulation of METTL3 inhibited LPS-induced apoptosis. METTL3 altered apoptosis through the JAK2/STAT3 signaling pathway, which was modulated by EZH2 [29]. The expression and apoptosis of METTL3 in alveolar epithelial cells of mice treated with LPS increased, and METTL3 overexpression exacerbated LPS-induced apoptosis, while knockdown of METTL3 alleviated cell apoptosis. This apoptosis mechanism is due to the interaction between METTL3 and neprilysin, which affects cell apoptosis [30]. In METTL3-silenced HepG2 cells, the METTL3 knockdown group and the control group showed differential gene expression profiles, where 22 out of 23 genes in the P53 signaling pathway had differentially expressed splice variants, and 18 genes were methylated. It was speculated that splicing of the P53 signaling pathway is associated with apoptosis induced by silencing METTL3 [31]. These findings indicate that METTL3 is related to apoptosis. However, there are differing explanations for the effect of METTL3 on apoptosis. In osteoblasts stimulated by LPS differentiation, the expression level of METTL3 is downregulated, and METTL3 knockdown leads to a higher apoptosis rate in LPS-induced osteoblasts. The expression of the anti-apoptotic protein BCL-2 decreased, while the apoptotic proteins cleaved Caspase-3, cleaved PARP-1, and cleaved Caspase-12 increased. Mechanistically, METTL3 knockdown enhanced the expression of Grp78 and affected cell apoptosis through YTHDF2-mediated RNA degradation [32].

Low levels of ROS can be beneficial, but excessive accumulation can have adverse effects. ROS regulate several cell signaling pathways involved in cellular transformation, inflammation, angiogenesis, and cancer, primarily through protein pathways such as transcription factors NF-κB and STAT3, hypoxia-inducible factor-1α, kinases, growth factors, cytokines, and others [33]. ROS are regulated by an antioxidant defense network that enables them to play a physiological role in minimizing ROS-induced oxidative damage, which can lead to disease. This network consists of various antioxidant scavenging enzymes such as Mn-SOD and CuZn-SOD [34]. In an in vitro study, human lens epithelial cells were exposed to high glucose conditions to simulate diabetic cataract conditions. It was found that METTL3 mediates the maturation of pri-miR-4654, thereby increasing the content of miR-4654 in the cells. MiR-4654 targets SOD2, thereby affecting apoptosis and oxidative stress in human lens epithelial cells [35]. Additionally, m6A mediates arsenic-induced oxidative stress and promotes the formation and maintenance of arsenic carcinogenic abnormal redox homeostasis. METTL3 upregulates m6A and promotes the expression of five key antioxidant enzymes (including SOD1, SOD2, TXN, CAT, and GPX1), which are involved in the regulation of oxidative stress and antioxidant balance [36]. The results of this study suggest that interference with METTL3 reduces LPS-induced ROS levels and upregulates the expression of the antioxidant enzymes *CAT* and *Mn-SOD*. Overexpression of METTL3 reverses this effect, suggesting that low expression of METTL3 may reduce, to some extent, the oxidative damage induced by LPS in sheep IECs.

The results of this study found that low-level METTL3 expression can alleviate LPS-induced damage to IECs to a certain extent, indicating that METTL3 may be a candidate gene for resistance to bacterial microbial infection. Although the function of METTL3 in LPS-induced IECs has been initially characterized in vitro, the identification of key mutation sites in the METTL3 gene through large-scale population association analysis remains a prerequisite for applying these findings to sheep disease resistance breeding. Candidate gene-based association analysis for marker-assisted selection (MAS) in livestock disease resistance faces potential challenges, such as acquiring large-scale populations of resistant and susceptible individuals and defining clear criteria for resistance and susceptibility. In recent years, new methods have emerged in the field of breeding, such as using the new gene-editing tool IS-Dra2-TnpB to target and integrate the screened inflammatory regulatory sequence (IRS) into the anti-mastitis lysozyme gene (LYZ) and using somatic cell cloning technology to obtain gene-edited dairy goat individuals. Both in vivo and in vitro experiments showed that the ability of gene-edited dairy goats to resist mastitis was significantly improved under bacterial infection conditions [37]. This breeding strategy based on gene-editing technology points to a direction for the application of the METTL3 gene in sheep disease resistance breeding.

## 4. Materials and Methods

### 4.1. Immunofluorescence Identification of Sheep IECs

IECs were isolated from the intestinal tissue of healthy newborn lambs, purified, and identified using the epithelial cell marker cytokeratin 18 antibody (Affinity Biosciences, Cincinnati, OH, USA, 1:500) [38,39]. For immunofluorescence experiments, IECs were inoculated into 24-well cell plates and washed with PBS (Solarbio, Beijing, China) when they reached the appropriate density. The samples were fixed with 4% paraformaldehyde (Solarbio, Beijing, China) for 30 min, and after fixation, the IECs were permeabilized with 0.5% Triton X-100 (Solarbio, Beijing, China) for 15 min and then washed with PBS. A 5% BSA solution (Solarbio, Beijing, China) was used for blocking at 37 °C for 30 min. The samples were then incubated with the primary antibody at 4 °C in the dark for 12 h. After incubation, the cells were washed with PBS and incubated with the corresponding secondary antibody at room temperature in the dark for 1 h. The nuclei of IECs were stained with DAPI (Beyotime, Shanghai, China). To observe the staining, the cells were examined and photographed using a fluorescence inverted microscope (Nikon, Tokyo, Japan). The primary antibody used was anti-CK18 (Affinity Biosciences, Cincinnati, OH, USA, 1:500), and the secondary antibody was Goat Anti-Rabbit IgG H&L (Abcam, Cambridge, UK, 1:500).

### 4.2. CCK-8 Assay

IECs were cultured in 96-well cell culture plates until they reached approximately 80% confluency. LPS from *Escherichia coli* 055 (Sigma-Aldrich, St. Louis, MO, USA) was then added at concentrations of 0 μg/mL, 0.5 μg/mL, 1 μg/mL, 2 μg/mL, 5 μg/mL, and 10 μg/mL. According to the instructions of the CCK-8 Kit (Vazyme, Nanjing, China), cell viability was assayed after 4 h. The absorbance at 450 nm was detected using a multi-mode microplate detection system (EnSpire, PerkinElmer, Waltham, MA, USA).

### 4.3. Western Blot

The IECs were lysed using RIPA lysis buffer (Beyotime, Shanghai, China) combined with protease inhibitor PMSF (Beyotime, Shanghai, China). The lysed cells were then centrifuged at 12,000 rpm for 10 min at 4 °C, and the resulting supernatant was collected. Protein concentration was determined using a BCA protein quantification kit (Beyotime, Shanghai, China). The protein samples were denatured and subjected to electrophoresis, followed by membrane transfer. The membrane was blocked with 5% skimmed milk powder at room temperature for 1 h. Primary antibodies were incubated with the membrane overnight at 4 °C, followed by a 1.5 h incubation with the secondary antibodies. TBST (Solarbio, Beijing, China) was used for washing after each antibody incubation, and the ECL chromogenic solution (Vazyme, Nanjing, China) was used for detection. The images were captured using a ChemDoc™ Touch Imaging System (Bio-Rad, Hercules, CA, USA). The primary antibodies and their dilution ratios were as follows: Anti-METTL3 antibody (Abmart, Shanghai, China, 1:2000) and Anti-beta Actin antibody (Affinity Biosciences, Cincinnati, OH, USA, 1:5000). The secondary antibodies and their dilution ratios were as follows: Goat Anti-Rabbit IgG H&L (HRP) (Abcam, Cambridge, UK, 1:5000) and Rabbit Anti-Mouse IgG H&L (HRP) (Abcam, Cambridge, UK, 1:5000).

### 4.4. Total RNA Extraction, cDNA Synthesis and qRT-PCR

Total RNA was extracted from primary IECs using Trizol (Takara, Dalian, China) and stored at −80 °C. RNA concentration was determined using an ultra-micro-volume UV/VIS spectrophotometer (Life Real, Hangzhou, China), confirming that the RNA was well preserved and met the requirements for subsequent experiments. cDNA was synthesized using a reverse transcription kit (Tiagen, Beijing, China), and gene expression levels were detected using the 2× TSINGKE^®^ Master qPCR Mix (Tsingke, Nanjing, China). qRT-PCR was conducted using a real-time fluorescence quantifier (Bio-Rad, Hercules, CA, USA). The qRT-PCR gene primers used in this study were designed using Premier Primer 5.0 software (Premier Biosoft International, Palo Alto, CA, USA) and synthesized by TsingKe (Nanjing, China). The list of primers is shown in Table 1.

### 4.5. Construction of Overexpression Vectors and Interfering Sequences

Table 2 shows the primer sequences used for homologous recombination in constructing the METTL3 overexpression vector. The CDS region of METTL3 was amplified using homologous recombination primers, and the pcDNA3.1 empty vector was enzymatically cleaved to obtain a linear vector. The CDS region of METTL3 was then inserted into the MCS region of the pcDNA3.1 empty vector using the ClonExpress II One Step Cloning Kit (Vazyme, Nanjing, China), resulting in the METTL3 overexpression vector. The overexpression vector used was the pcDNA3.1 plasmid (Genepharma, Suzhou, China), and the fluorescent vector used was the pcDNA3.1-EGFP (Genepharma, Suzhou, China). Table 3 shows the METTL3 interference sequences designed and synthesized by Genepharma, with the FAM-labeled negative control also provided by Genepharma.

### 4.6. Cell Culture and Cell Transfection

Well-grown IECs were inoculated into cell culture plates with DMEM-F12 medium (Sigma-Aldrich, St. Louis, MO, USA) containing 10% fetal bovine serum (Gibco, Grand Island, NY, USA) and 1% penicillin-streptomycin-amphotericin (Solarbio, Beijing, China). All IECs were cultured in a cell incubator (Thermo, Waltham, MA, USA) at 37 °C with 5% CO2. IECs were transfected at approximately 50% confluency, and cells were collected after 36 h. The jetPRIME Transfection Reagent (Polyplus, Illkirch, France) was used for cell transfection. All cell transfection procedures were performed in accordance with the manufacturer’s protocol.

### 4.7. ELISA Assay

Remove the cell culture and centrifuge at 1000 rpm for 20 min. Collect the supernatant and store it at −80 °C for later use. The experimental procedures in this study were performed according to the instructions of the sheep IL1β ELISA kit (mIbio, Shanghai, China), sheep IL6 ELISA kit (mIbio, Shanghai, China), and sheep TNFα ELISA kit (mIbio, Shanghai, China). Standard curves for IL1β, IL6, and TNFα were generated using the protein standards provided in the kits to obtain the OD values at different concentrations, which were then used to plot the standard curves (Figure 8).

### 4.8. Apoptosis Detection

The kit used to detect apoptosis in IECs cells was Annexin V-FITC/PI Apoptosis Detection Kit (Vazyme, Nanjing, China), and Annexin V-FITC and PI were labelled for early apoptosis and late apoptosis, respectively, and the specific procedure was carried out according to the instruction manual, while the detection was performed on the machine after processing.

### 4.9. Reactive Oxygen Detection

2′,7′-Dichlorodihydrofluorescein diacetate (DCFH-DA) is a versatile indicator of oxidative stress. DCFH-DA was diluted with a serum-free culture medium to a final concentration of 10 μM, using a 1:1000 ratio. Cells were suspended in the diluted DCFH-DA at a concentration of 1–20 million cells per mL and incubated in a 37 °C cell culture incubator for 20 min. The cells were mixed by inverting the container every 3–5 min to ensure full contact with the probe. After incubation, the cells were washed three times with serum-free culture medium to remove any remaining DCFH-DA that did not enter the cells. The experiment involved testing the cells using a reactive oxygen species detection kit (Beyotime, Shanghai, China).

### 4.10. Statistical Analysis

SPSS 25.0 software (SPSS Inc., Chicago, IL, USA) was used for statistical analysis. The unpaired Student’s *t*-test was used for the two-group comparison analysis. The data were considered statistically significant only when *p* < 0.05 (*), *p* < 0.01 (**), or *p* < 0.001 (***). Each experiment included three biological replications. All experiment data are reported as means ± SEM (standard error of the mean).

## 5. Conclusions

In this study, immunofluorescence identification confirmed that the extracted IECs were epithelial cells. LPS induced an inflammatory response, apoptosis, and oxidative stress in IECs, and the protein expression level of METTL3 was upregulated during LPS-induced IECs. By overexpressing and knocking down METTL3 in IECs, the results showed that low-level expression of METTL3 could reduce the LPS-induced inflammatory response, apoptosis, and oxidative stress in IECs. These findings suggest that METTL3 may be a potential genetic marker for resistance to cellular damage.

## Figures and Tables

**Figure 1 ijms-25-09316-f001:**
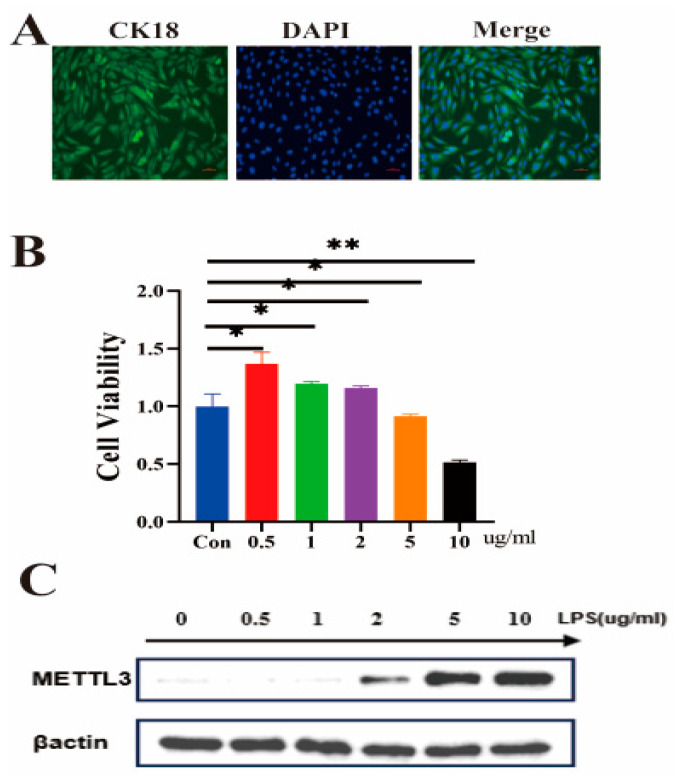
Immunofluorescence analysis of the epithelial cell marker CK18 on IECs. (**A**) CCK8 detection results after 4 h of treatment with LPS at concentrations of 0 μg/mL, 0.5 μg/mL, 1 μg/mL, 2 μg/mL, 5 μg/mL, and 10 μg/mL on IECs (**B**). Protein expression of METTL3 in IECs after 4 h of treatment with the same LPS concentrations (**C**). The image scale is 200×. The data are presented as means ± SEM (standard error of the mean) (*n* = 3). The statistical significance was assessed using the unpaired Student’s *t*-test. (Unmarked: *p* > 0.05; * *p* < 0.05; ** *p* < 0.01).

**Figure 2 ijms-25-09316-f002:**
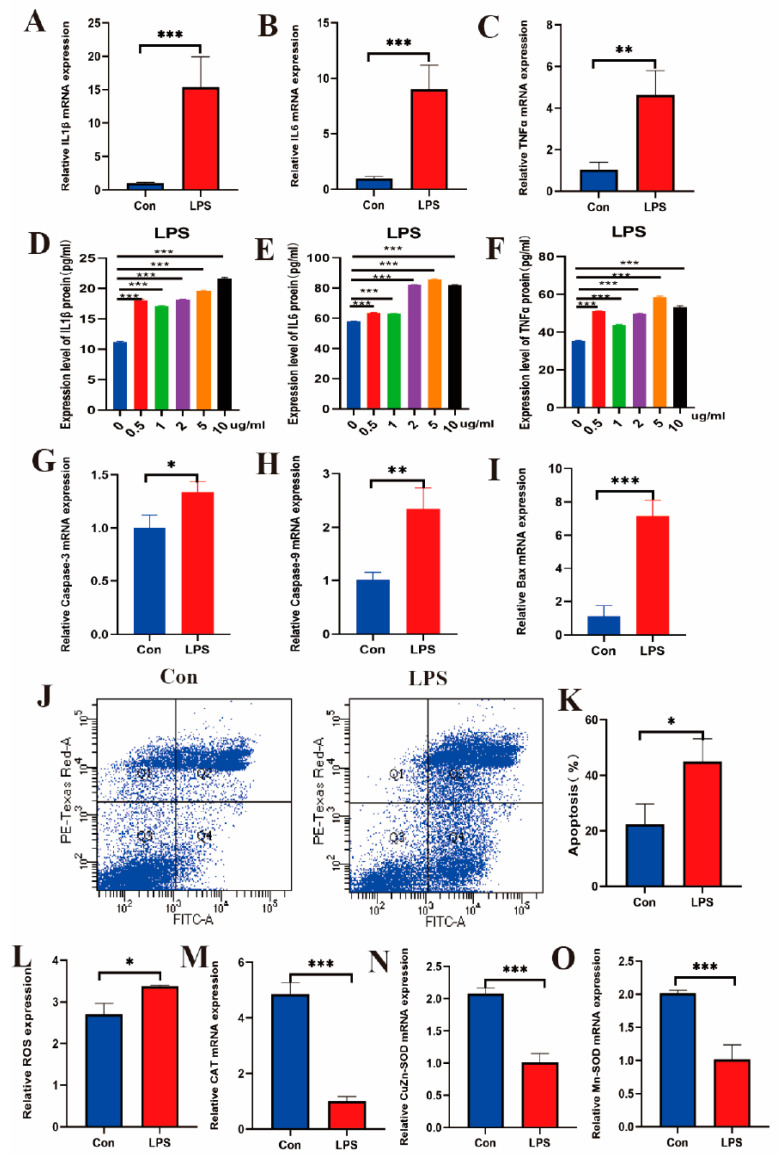
The mRNA levels of cytokines *IL1β*, *IL6*, and *TNFα* in IECs in the LPS group and control group (**A**–**C**). Protein levels of IL1β, IL6, and TNFα in IECs cell cultures stimulated by different concentrations of LPS (**D**–**F**). The mRNA levels of apoptosis marker genes *caspase-3*, *caspase-9*, and *Bax* in the LPS group and control group (**G**–**I**). Cells in different states in the LPS group and control group, with Q1, Q2, Q3, and Q4, respectively, representing necrotic or mechanically damaged cells, late apoptotic cells, living cells, and early apoptotic cells (**J**). Statistical analysis of the proportion of apoptotic cells in the LPS group and control group (**K**). ROS levels (**L**) and mRNA levels of antioxidant marker genes *CAT*, *Mn-SOD*, and *CuZn-SOD* in the LPS group and control group (**M**–**O**). The data are presented as means ± SEM (standard error of the mean) (*n* = 3). The statistical significance was assessed using the unpaired Student’s *t*-test. (Unmarked: *p* > 0.05; * *p* < 0.05; ** *p* < 0.01; *** *p* < 0.001).

**Figure 3 ijms-25-09316-f003:**
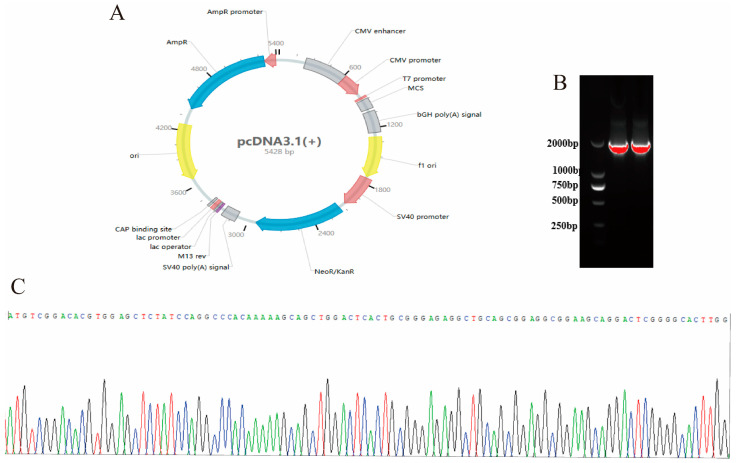
Legend for pcDNA3.1 empty vector (**A**). PCR identification of the *METTL3* overexpression vector (**B**). Sanger sequencing results of the *METTL3* overexpression vector (**C**).

**Figure 4 ijms-25-09316-f004:**
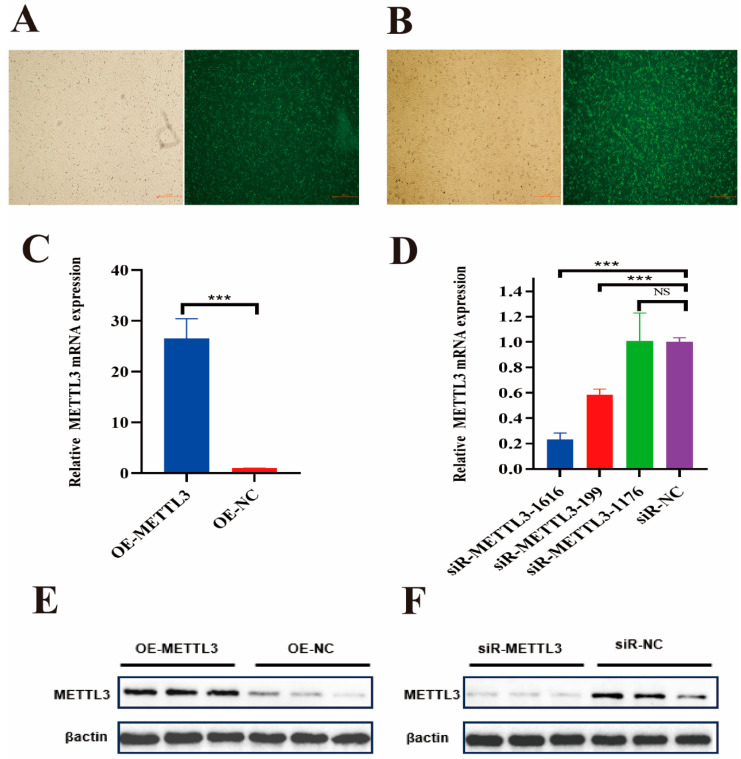
Fluorescence images of pcDNA3.1-EGFP fluorescent vector and FAM-labelled negative control in IECs. (**A**,**B**). The mRNA levels after *METTL3* overexpression and interference in IECs. (**C**,**D**). Expression of protein level after overexpression and interference of METTL3 in IECs. (**E**,**F**). The image scale is 100×. The data are presented as means ± SEM (standard error of the mean) (*n* = 3). The statistical significance was assessed using the unpaired Student’s *t*-test. (NS: *p* > 0.05; *** *p* < 0.001).

**Figure 5 ijms-25-09316-f005:**
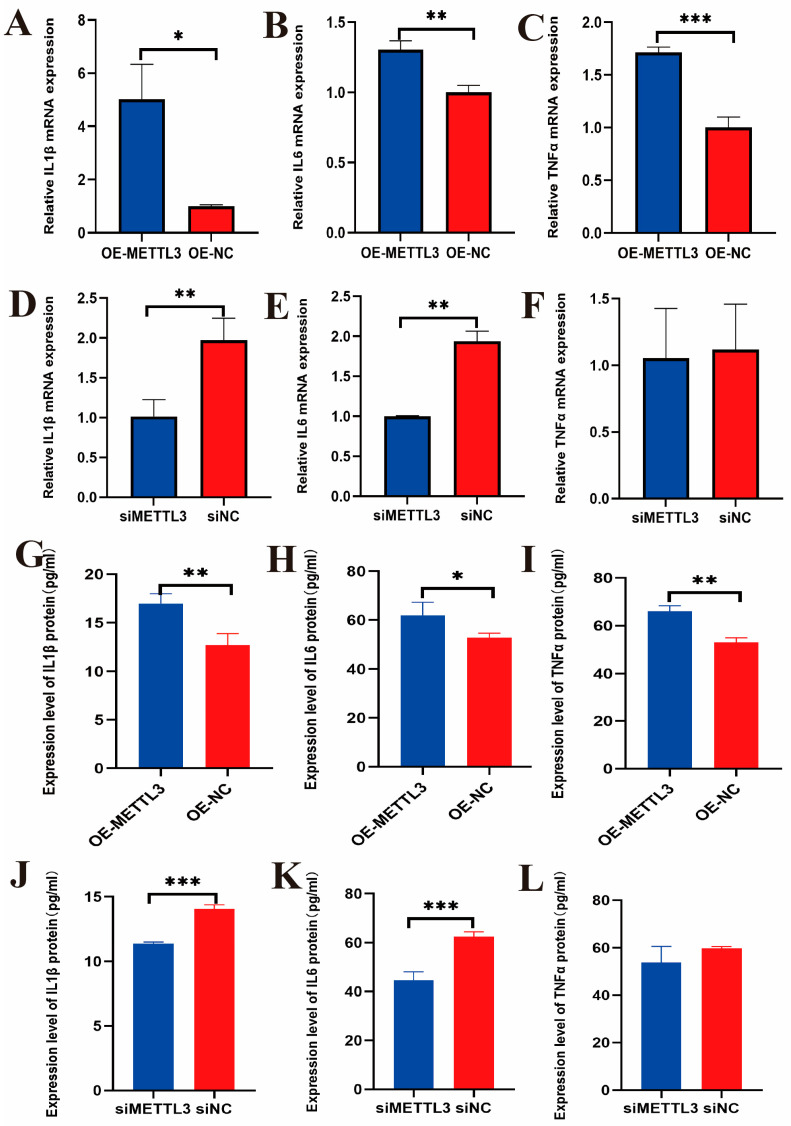
The mRNA expression levels of cytokines *IL1β*, *IL6*, and *TNFα* in IECs after LPS induced overexpression of *METTL3* (**A**–**C**). Protein expression levels of cytokines IL1β, IL6, and TNFα in IECs after LPS induced overexpression of *METTL3* (**D**–**F**). The mRNA expression levels of cytokines IL1β, IL6, and TNFα in IECs after LPS induced interference with *METTL3* (**G**–**I**). Protein expression levels of cytokines IL1β, IL6, and TNFα in IECs after LPS induced interference with *METTL3* (**J**–**L**). The data are presented as means ± SEM (standard error of the mean) (*n* = 3). The statistical significance was assessed using the unpaired Student’s *t*-test. (Unmarked: *p* > 0.05; * *p* < 0.05; ** *p* < 0.01; *** *p* < 0.001).

**Figure 6 ijms-25-09316-f006:**
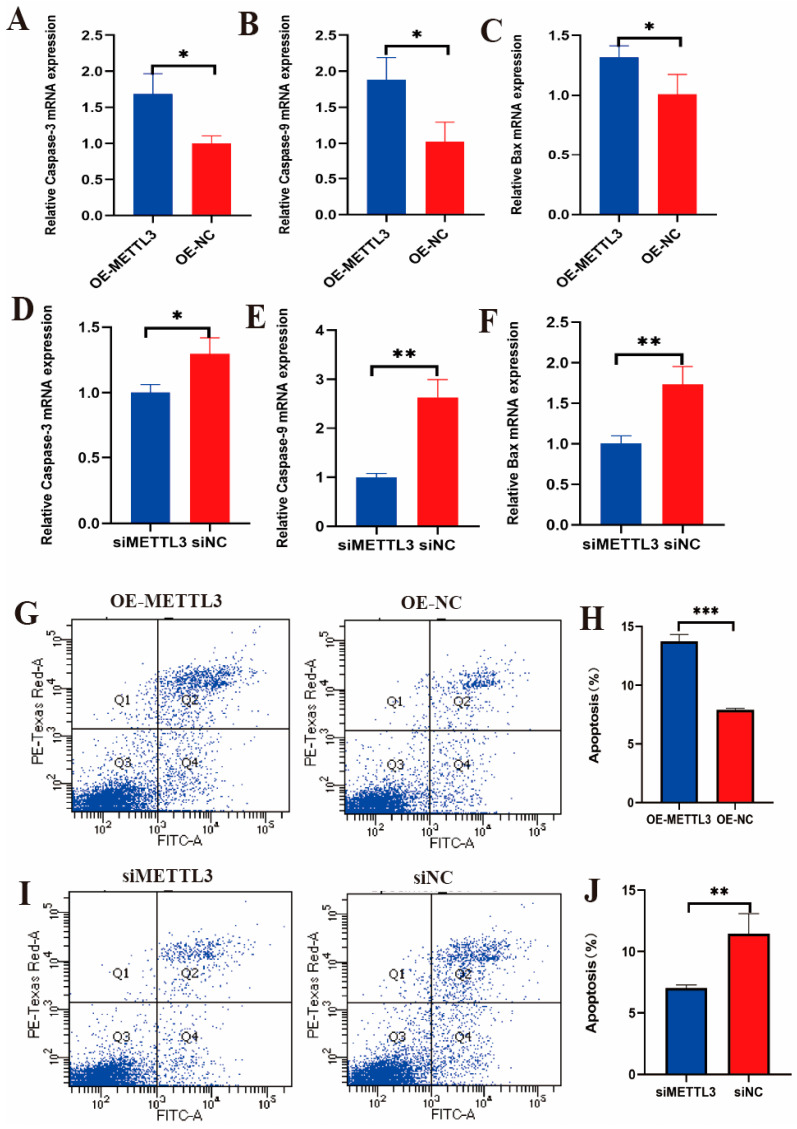
The mRNA levels of apoptosis marker genes *Caspase-3*, *Caspase-9*, and *Bax* in IECs after LPS-induced *METTL3* overexpression (**A**–**C**). The mRNA levels of apoptosis marker genes *Caspase-3*, *Caspase-9*, and *Bax* in IECs after LPS-induced interference with *METTL3* (**D**–**F**). Cells in different states of IECs after LPS-induced overexpression and interference of *METTL3*, with Q1, Q2, Q3, and Q4, respectively, representing necrotic or mechanically damaged cells, late apoptotic cells, living cells, and early apoptotic cells (**G**,**I**). Statistical analysis of apoptotic cells in IECs after LPS-induced overexpression and interference of *METTL3* (**H**,**J**). The data are presented as means ± SEM (standard error of the mean) (*n* = 3). The statistical significance was assessed using the unpaired Student’s *t*-test. (Unmarked: *p* > 0.05; * *p* < 0.05; ** *p* < 0.01; *** *p* < 0.001).

**Figure 7 ijms-25-09316-f007:**
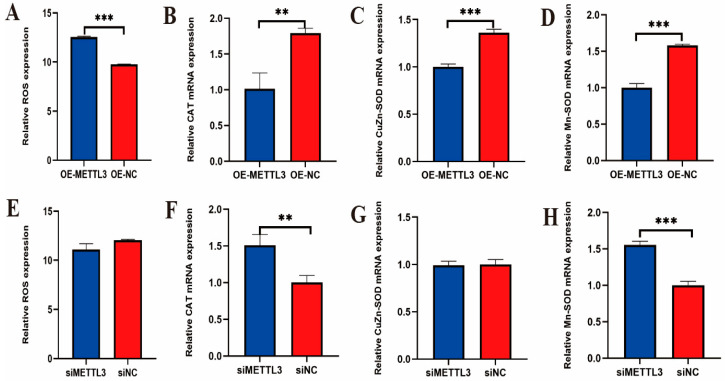
ROS levels in IECs after LPS-induced overexpression of *METTL3* (**A**) and ROS levels in IECs after LPS-induced interference with *METTL3* (**E**). The mRNA expression levels of antioxidant marker genes *CAT*, *Mn-SOD*, and *CuZn-SOD* in IECs after LPS-induced overexpression of *METTL3* (**B**–**D**) and the mRNA expression levels of antioxidant marker genes *CAT*, *Mn-SOD*, and *CuZn-SOD* in IECs after LPS-induced interference with METTL3 (**F**–**H**). The data are presented as means ± SEM (standard error of the mean) (*n* = 3). The statistical significance was assessed using the unpaired Student’s *t*-test. (Unmarked: *p* > 0.05; ** *p* < 0.01; *** *p* < 0.001).

**Figure 8 ijms-25-09316-f008:**
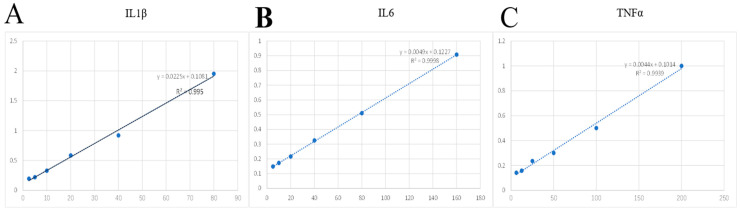
(**A**–**C**) represent the standard curves for IL1β, IL6, and TNFα, respectively, with the X-axis representing the different concentrations of standards (pg/mL) and the Y-axis representing the OD values obtained for each concentration.

**Table 1 ijms-25-09316-t001:** Gene primers used for qRT-PCR.

Gene	Primer Sequence (5′-3′)	Product Size (bp)	Annealing Temperature (°C)
*METTL3*	F: CCCTATGGGACCCTGACAGA R: CCTGTTCGAATGATGCGCTG	197	60
*β-actin*	F: CAGTCGGTTGGATCGAGCAT R: AGAAGGAGGGTGGCTTTTGG	151	60
*IL1β*	F: CGTGCCTACGAACATGTC R: CACCAGGGATTTTTGCTCTC	168	60
*IL6*	F: GACTTCTGCTTTCCCTACCC R: CACACTCGTCATTCTTCTCAC	171	60
*TNFα*	F: CTCGTATGCCAATGCCCTCA R: TGGTGTGGGTGAGGAACAAG	149	60
*Caspase-3*	F: ACGTTGTGGCTGAACGTAAA R: GTTTCCCTGAGGTTTGCTGC	262	60
*Caspase-9*	F: CAACGTGAACTTCTGCCGTG R: CATTTGCTTGGCAGTCAGGT	136	60
*Bax*	F: GCCCTTTTGCTTCAGGGTTT R: GTCCAATGTCCAGCCCATGA	357	60
*CAT*	F: CTATCCTGACACTCACCGCC R: CTTTCAGATGGCCCGCAATG	335	60
*CuZn-SOD*	F: TGATCATGGGTTCCACGTCC R: CACATTGCCCAGGTCTCCAA	139	60
*Mn-SOD*	F: GGATCCCCTGCAAGGAACAA R: CTTGGTGTAAGGCTGACGGT	189	60

**Table 2 ijms-25-09316-t002:** The METTL3 overexpression vector was used to construct homologous recombination primer sequences.

Primer Name	Primer Sequence (5′-3′)	Product Size (bp)	Annealing Temperature (°C)
OE-METTL3	F:ctagcgtttaaacttaagcttATGTCGGACACGTGGAGCTC R:tgctggatatctgcagaattcCTATAGATTCTTAGGTTTAGAGATGATACCAT	1782	62

**Table 3 ijms-25-09316-t003:** METTL3 interference sequence.

Fragment Name	Sequence (5′-3′)
SiR-METTL3-1616	F:GCUGAGGUUCGUUCCACUATT R:UAGUGGAACGAACCUCAGCTT
SiR-METTL3-199	F: GCACUUGGAUCUUCGGAAUTT R: AUUCCGAAGAUCCAAGUGCTT
SiR-METTL3-1176	F:GCGUUGGAGGUGACUCCAATT R:UUGGAGUCACCUCCAACGCTT
SiR-NC	F:UUCUCCGAACGUGUCACGUTT R:ACGUGACACGUUCGGAGAATT

## Data Availability

The data analyzed in this study are available from the corresponding author upon request.

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
