# Peer review of "Effect of METTL3 Gene on Lipopolysaccharide Induced Damage to Primary Small Intestinal Epithelial Cells in Sheep"

_ijms, 2024, doi:10.3390/ijms25179316_

Round 1

Reviewer 1 Report

Comments and Suggestions for Authors

- Results headings need to be revised. 

- Should avoid using words like "extremely significantly" (Line 80). There are several more throughout the manuscript.

- Provide raw blots of all Western blot images used in the manuscript. 

- All figures need to be re-sized; fonts are too small. Sometimes, it's difficult to read.

- Fig 1B: why did LPS at 0.5 ug/ml increased cell viability when METTL3 did not change? Also, why do control cells have 0.6 viability, not 1.0?

- Fig 2I: missing asterisk

- Fig 2J, 5G, I: flow plots need more information and look not clean. More information needs to be stated in the figure legends.

- Fig 3D,E: hard to see

- Discussion should be expanded to thoroughly support the data.

Overall, the manuscript, at this current format, needs a major revision.

Comments on the Quality of English Language

- The manuscript should be taken a closer look as there are typos, spacing errors, and few sentences are too long (can be separated for clarity). 

Author Response

Comments 1:Results headings need to be revised.

Response 1:The titles of all results sections have been revised to better align with the content of the article.

Comments 2: Should avoid using words like "extremely significantly" (Line 80). There are several more throughout the manuscript.

Response 2:The full text has been revised to remove terms like "extremely significant."

Comments 3: Provide raw blots of all Western blot images used in the manuscript. 

Response 3:The original data of Westen Blot was provided when submitting the manuscript.  The editor may have forgotten to provide it to the reviewer, so I will provide it to you in the attachment.

Comments 4: All figures need to be re-sized; fonts are too small. Sometimes, it's difficult to read.

Response 4:The drawings have been redone, and the numbers in all images have been resized for easier readability.

Comments 5:Fig 1B: why did LPS at 0.5 ug/ml increased cell viability when METTL3 did not change? Also, why do control cells have 0.6 viability, not 1.0?

Response 5:IECs exhibit a dose- and time-dependent response to LPS; if the exposure time is too short or the dose is too low, the impact on the cells may be minimal. While the cells are not affected by low doses or short-term exposure to LPS, they continue to proliferate, leading to a certain degree of upregulation in cell activity. The protein levels of METTL3 also do not change until the cells are exposed to a sufficient concentration of LPS or for a sufficient duration. Only then might changes in cell activity and METTL3 protein levels become apparent.

 why do control cells have 0.6 viability, not 1.0?

The reason the viability of the control cells initially showed 0.6 instead of 1.0 is due to an error in the calculation of cell viability. The correct formula for calculating cell viability is:

Cell viability % = (medicated cells OD-blank OD)/(control cells OD-blank OD) × 100%.However, in my initial calculation, I did not subtract the OD of the blank group from the OD values of the medicated and control groups. This omission led to an incorrect viability calculation, where the control group did not show the expected 1.0 (i.e., 100%). After discovering this error, I corrected the calculation by properly subtracting the blank OD, which should now yield accurate viability percentages, with the control cells showing the expected 1.0.

Comments 6: Fig 2I: missing asterisk

Response 6:Modifications have been made and the modification location is still in Fig2I.

Comments 7:Fig 2J, 5G, I: flow plots need more information and look not clean. More information needs to be stated in the figure legends.

Response 7:The flow diagram in Figure 2J has been adjusted for clarity, and the added annotation is now located in lines 122-123. Similarly, the flow diagrams in Figures 5G and 5I have also been adjusted for clarity, with annotations added in lines 200-202. These diagrams are now referred to as Figures 6G and 6I.

Comments 8: Fig 3D,E: hard to see

Response 8:Figures 3D and 3E have been adjusted and are now positioned as Figures 4A and 4B.

Comments 9: Discussion should be expanded to thoroughly support the data.

Response 9:The Discussion section has been largely rewritten, incorporating additional literature and content. The specific modifications are now located in lines 236-249, 262-274, 289-297, and 302-318.

Comments 10:The manuscript should be taken a closer look as there are typos, spacing errors, and few sentences are too long (can be separated for clarity). 

Response 10:The entire article has been refined.

Reviewer 2 Report

Comments and Suggestions for Authors

This paper explores the effect of METTL3 on LPS-induced damage in sheep intestinal epithelial cells (IECs). The study is well-structured and provides valuable insights into the role of m6A modifications in regulating inflammatory responses and apoptosis. However, there are several areas that need improvement in terms of clarity, methodological detail, and scientific rigor.

Introduction is too brief. For istance, Line 38-42: The description of the innate immune system's role is somewhat general. Provide more specific background on how LPS interacts with TLR4 in sheep IECs and how this interaction might differ from other species.

Discussion is approached a little too hastily.

Examples Line 136-138: The discussion on the role of METTL3 in promoting inflammatory responses is sound but could explore potential molecular mechanisms in greater detail.

Please, discuss how METTL3 might interact with other known regulators of inflammation and apoptosis, such as NF-κB or p53.

Line 156-157: The implications of METTL3 in disease resistance breeding are mentioned but not fully developed.

Elaborate on how these findings could be applied in practical breeding programs, including potential challenges.

Last but not least, the images need to be completely reorganized. For example, figure 1B appears out of shape and the asterisks are not clearly visible. In figure 2 I suggest putting the letters to the left of the various graphs.

Figure 3 really contains a lot of results and the immunofluorescence images are so small that it is impossible to fully appreciate them; not having so many images in the manuscript this figure could be divided into two figures. Figures 4-5-6 see commentary on figure 2.

Percentage match of plagiarism is quite high (34%)

Comments on the Quality of English Language

Minor editing of English language required.

Author Response

Comments 1: Introduction is too brief. For istance, Line 38-42: The description of the innate immune system's role is somewhat general. Provide more specific background on how LPS interacts with TLR4 in sheep IECs and how this interaction might differ from other species.

Response 1:As requested, the introduction now includes background information on LPS and TLR4, as well as the current research status in other species and sheep. Now lines 38-60.

Comments 2: Discussion is approached a little too hastily.

Examples Line 136-138: The discussion on the role of METTL3 in promoting inflammatory responses is sound but could explore potential molecular mechanisms in greater detail.

Please, discuss how METTL3 might interact with other known regulators of inflammation and apoptosis, such as NF-κB or p53.

Response 2:The Discussion section now provides a more detailed explanation of the potential mechanisms by which METTL3 affects inflammatory responses, specifically in lines 236-256. Additionally, the underlying mechanism regulated by the inflammatory factor NF-κB is detailed in lines 241-251. Because P53 plays a crucial role in the cell apoptosis pathway, the discussion of METTL3's influence on cell phenotype through P53 has been included in the apoptosis section, now located in lines 269-274.

Comments 3:  Line 156-157: The implications of METTL3 in disease resistance breeding are mentioned but not fully developed.

Elaborate on how these findings could be applied in practical breeding programs, including potential challenges.

Response 3:The Discussion section has been expanded to address the significance of METTL3 in disease resistance breeding, including how it can be implemented and the challenges involved. This content is now located in lines 302-318.

Comments 4: Last but not least, the images need to be completely reorganized. For example, figure 1B appears out of shape and the asterisks are not clearly visible. In figure 2 I suggest putting the letters to the left of the various graphs.

Response 4:Figures 1 and 2 have been redrawn as requested.

Comments 5:  Figure 3 really contains a lot of results and the immunofluorescence images are so small that it is impossible to fully appreciate them; not having so many images in the manuscript this figure could be divided into two figures. Figures 4-5-6 see commentary on figure 2.

Response 5:The clarity of all images has been enhanced, Figure 3 has been split into two separate images as requested, and the annotations for all figures have been improved.

Comments 6: Percentage match of plagiarism is quite high (34%)

Response 6:When writing the first draft, many of the experimental methods were similar, and some were based on the previous methods used by our research group, leading to a high duplication rate. Software has since been used to reduce this duplication.

Comments 7: Minor editing of English language required.

Response 7:The entire article has been thoroughly polished.

Round 2

Reviewer 1 Report

Comments and Suggestions for Authors

Authors have addressed all of my comments.

Reviewer 2 Report

Comments and Suggestions for Authors

The authors well replied to my previous comments, nothing to add.

Comments on the Quality of English Language

None